# Exosomes in Skin Flap Survival: Unlocking Their Role in Angiogenesis and Tissue Regeneration

**DOI:** 10.3390/biomedicines13020353

**Published:** 2025-02-04

**Authors:** Bo-da Chen, Yue Zhao, Jian-long Wu, Zi-guan Zhu, Xiao-dong Yang, Ren-peng Fang, Chen-si Wu, Wei Zheng, Cheng-an Xu, Keyang Xu, Xin Ji

**Affiliations:** 1Center for Plastic & Reconstructive Surgery, Department of Hand & Reconstructive Surgery, Zhejiang Provincial People’s Hospital (Affiliated People’s Hospital), Hangzhou Medical College, Hangzhou 310014, China; chenboda@hmc.edu.cn (B.-d.C.); wujianlong@hmc.edu.cn (J.-l.W.); zhuziguan@hmc.edu.cn (Z.-g.Z.); yangxdyw@163.com (X.-d.Y.); fangrenpeng@hmc.edu.cn (R.-p.F.); 2School of Public Health, Hangzhou Medical College, Hangzhou 310053, China; zhaoy0701@163.com; 3Center for General Practice Medicine, Department of Infectious Diseases, Zhejiang Provincial People’s Hospital (Affiliated People’s Hospital), Hangzhou Medical College, Hangzhou 310014, China; wuchensi@zju.edu.cn (C.-s.W.); zw19931125@126.com (W.Z.); 13588172730@163.com (C.-a.X.); 4State Key Laboratory of Quality Research in Chinese Medicine, Faculty of Chinese Medicine, Macau University of Science and Technology, Macau 999078, China; 21482055@life.hkbu.edu.hk

**Keywords:** exosomes, skin flap survival, angiogenesis, tissue regeneration, mechanism

## Abstract

This review explores the critical role of exosomes in promoting angiogenesis, a key factor in skin flap survival. Skin flaps are widely used in reconstructive surgery, and their survival depends heavily on the formation of new blood vessels. Exosomes, small extracellular vesicles secreted by various cells, have emerged as important mediators of intercellular communication and play a crucial role in biological processes such as angiogenesis. Compared to traditional methods of promoting angiogenesis, exosomes show more selective and targeted therapeutic potential as they naturally carry angiogenic factors and can precisely regulate the angiogenesis process. The review will delve into the molecular mechanisms by which exosomes facilitate angiogenesis, discuss their potential therapeutic applications in enhancing skin flap survival, and explore future research directions, particularly the challenges and prospects of exosomes in clinical translation. By highlighting the unique advantages of exosomes in skin flap survival, this review provides a new perspective in this field and opens up new research directions for future therapeutic strategies.

## 1. Introduction

Skin flap surgery is a crucial technique in plastic and reconstructive surgery, employed to repair tissue defects by transferring a segment of tissue, along with its blood supply, from one area of the body to another [1]. This procedure is vital for restoring both function and aesthetics in regions affected by trauma, disease, or congenital defects. Unlike skin grafts, skin flaps consist not only of skin but also underlying tissues, such as fat, muscle, or even bone, and they retain their own blood supply. Skin flaps can be categorized into local, regional, free, and pedicle flaps, depending on the method of tissue transfer and whether the blood supply remains intact throughout the procedure [2].

Flap survival is fundamental to the success of skin flap surgery, primarily relying on adequate blood supply. Insufficient blood flow can lead to ischemia, resulting in flap necrosis (tissue death), infection, and reconstruction failure [3]. The blood supply to the skin and its underlying tissues is critical for their survival and healing. Major distributing vessels supply blood through cutaneous perforators, which penetrate the skin from the fascia or subfascial spaces, and musculocutaneous perforators, which traverse from the muscle layer into the skin. This arrangement forms a complex network of blood vessels, including the dermal and subdermal plexuses [4]. The physiological role of this vascular network is to ensure that the skin and subcutaneous tissues receive adequate blood supply both at rest and during wound repair. In flap surgery, surgeons depend on these vascular networks to ensure sufficient blood flow, thereby maintaining the viability and survival of the flap throughout the healing process [5,6].

Given the critical role of blood vessels in flap survival, strategies to enhance angiogenesis—the formation of new blood vessels—are vital for improving surgical outcomes. One emerging area of interest is the potential of exosomes in promoting angiogenesis. Exosomes are small extracellular vesicles, typically 30–100 nm in size, with a lipid bilayer structure, secreted by various cell types [7]. These vesicles are naturally present in body fluids, including blood, saliva, urine, cerebrospinal fluid, and breast milk [8]. They are involved in intercellular communication by transporting a variety of cargo, including proteins, lipids, and nucleic acids, that can influence the behavior of recipient cells. This includes the modulation of angiogenic factors, making exosomes particularly relevant to skin flap survival. Exosomes have been shown to contain pro-angiogenic factors such as vascular endothelial growth factor (VEGF) and fibroblast growth factor (FGF), which can directly stimulate endothelial cell proliferation and migration, promoting the formation of new blood vessels [9]. In pathological conditions such as cancer, cardiovascular diseases, and chronic inflammatory disorders, exosomes can significantly enhance angiogenesis, a key factor in tumor growth and metastasis [10]. Their content can also reflect the physiological state of the originating cell, positioning exosomes as non-invasive biomarkers for angiogenic activity in various diseases, thus aiding in early diagnosis and monitoring treatment responses [11]. Exosomes also hold considerable therapeutic potential. Targeting exosomes or manipulating their contents may offer novel strategies for enhancing or inhibiting angiogenesis, which could be beneficial in treating conditions like ischemia or preventing tumor growth [12]. Furthermore, in tissue engineering and regenerative medicine, exosomes derived from stem cells have demonstrated the ability to enhance angiogenesis and promote tissue repair, making them a promising tool for developing effective therapies [13].

In the context of skin flap surgery, exosomes could provide a novel therapeutic approach for enhancing flap survival by promoting angiogenesis and improving blood supply. Targeting exosomes to deliver angiogenic factors directly to the flap site or manipulating their contents to enhance vascularization may offer a valuable addition to current surgical techniques. This review will explore the molecular mechanisms through which exosomes promote angiogenesis, evaluate their potential therapeutic applications for improving skin flap survival, and outline prospective avenues for future research, particularly focusing on clinical translation and potential challenges in the application of exosomes in tissue regeneration.

## 2. Angiogenesis in Skin Flap Survival

### 2.1. The Physiological Process of Angiogenesis

Angiogenesis refers to the physiological process of forming new blood vessels based on pre-existing ones. It is an important process in growth, development, wound healing, and tissue injury response.

Following the process of vasculogenesis during development, angiogenesis creates new vascular structures through various mechanisms [14]. For instance, enhancing microvascular density through the sprouting of endothelial cells (ECs), increasing vascular surface area by vessel splitting via intussusceptive angiogenesis, and boosting blood flow through the fusion of capillaries in coalescent angiogenesis (Figure 1). Each mode contributes to the overall increase in vascular density and function, adapting to the specific needs of tissues in both physiological and pathological conditions [15,16].

Sprouting angiogenesis is considered the fastest method for vascularizing growing tissue, typically occurring in response to hypoxia, injury, or other signaling stimuli. In this process, ECs undergo a series of changes in the presence of VEGF, which promotes their proliferation and migration, and the degradation of the extracellular matrix (ECM) [14]. Although VEGF aids in the formation of new blood vessels, these newly formed vessels are initially immature and leaky; over time, they deposit a new ECM that attracts stabilizing pericytes [17]. During sprouting, specialized ECs explore their environment using structures called dactylopodia and filopodia, primarily regulated by VEGF and Neuropilin-1 (NRP1). Additionally, VEGF induces the formation of podosome rosettes, which degrade the basement membrane and facilitate both angiogenesis and anastomosis [18]. The selection of tip cells appears to be stochastic, with tip and stalk cells dynamically exchanging positions during the sprouting process [19,20]. Tip cells are enriched with various ECM components and genes associated with the transforming growth factor beta (TGFβ) pathway, and TGFβ signaling plays a crucial role in promoting sprouting [21]. A variant of angiogenesis distinct from sprouting is intussusceptive angiogenesis. This process was initially identified during the post-natal remodeling of lung capillaries, where existing vessels split into two new vessels following the formation of a trans-vascular pillar between two opposing ECs in the vessel lumen [22]. Intussusception is a rapid form of vascular remodeling that can occur in just hours or even minutes, as it does not initially rely on cell proliferation. Research has shown that pillar formation is not limited to capillary networks; it also takes place in smaller arteries and veins [23]. Blood vessels can remodel to form functional vascular trees from an initially homogeneous capillary mesh. This occurs in specific flow pathways where these pathways expand and fuse, while trans-vascular pillars are removed and less perfused capillaries regress. This type of angiogenesis, known as coalescent angiogenesis, is characterized by a reduction in the number of vessels but an increase in the diameter of the resulting vessels [24].

### 2.2. Factors Influencing Angiogenesis in Skin Flaps

Angiogenesis in skin flaps is a complex process that is influenced by a variety of factors and primarily driven by the need to maintain tissue viability. Hypoxia is a significant stimulus for angiogenesis as it triggers the activation of hypoxia-inducible factors (HIFs) [25], which then promote the expression of VEGF [26]. VEGF plays a pivotal role in both the normal and ischemic angiogenesis within skin flaps, increasing vascular permeability and promoting the proliferation and migration of ECs. A precise balance of VEGF is crucial, as even minor reductions can impair vessel formation, leading to inadequate blood supply to the skin flap.

Beyond VEGF, the ECM is equally important in supporting angiogenesis in skin flaps. It provides structural support and regulates EC migration and vessel formation. Remodeling of the ECM is facilitated by various proteinases, such as matrix metalloproteinases (MMPs), which break down ECM components and release bound angiogenic factors like VEGF and FGFs [27,28]. The degradation of the ECM is tightly regulated, and both excessive and insufficient proteolysis can disrupt angiogenesis [29]. Growth factors such as platelet-derived growth factor (PDGF) and angiopoietins (Ang-1 and Ang-2) are also involved in the stabilization and maturation of new vessels [30]. PDGF recruits pericytes and smooth muscle cells (SMCs) to the nascent vessels, which is essential for vessel stabilization. Ang-1 tightens EC junctions and enhances vessel stability, while Ang-2 loosens these junctions, allowing for EC migration and further vessel sprouting [31,32,33]. Leukocytes and inflammatory cells further influence angiogenesis by releasing cytokines and growth factors that promote or inhibit the process. For instance, monocytes and macrophages secrete VEGF and FGFs, which promote angiogenesis, while anti-inflammatory cytokines like interleukin-10 (IL-10) can inhibit it [34]. Additionally, the process of vessel regression, which occurs when blood supply exceeds tissue demand, is crucial to the balance of angiogenesis. Factors such as decreased perfusion or insufficient stabilization by pericytes can lead to vessel regression, which is often observed when skin flaps are exposed to ischemic conditions [35].

Understanding these factors is crucial for improving the success of skin flap surgeries by promoting proper vascularization and minimizing ischemia.

Mechanical stretch stimulates the production of angiogenic factors, enhancing local blood supply [36]. Additionally, preconditioning techniques, such as ischemic preconditioning (IPC), activate protective and pro-angiogenic signaling pathways by temporarily reducing blood flow, thus improving angiogenesis in skin flaps [3,37]. Stem cell therapy, particularly the use of mesenchymal stem cells (MSCs), supports the formation of new vascular networks by releasing growth factors [38]. Nitric oxide (NO), a vasodilator, not only enhances endothelial cell migration but also increases blood flow to ischemic tissues [39]. The interplay of these factors provides a crucial physiological basis for the healing and vascularization of skin flaps, and understanding these mechanisms is essential for improving the success rates of skin flap surgeries.

### 2.3. Advancements and Challenges in Promoting Angiogenesis for Skin Flap Survival

Surgical delay and IPC are two preconditioning techniques that enhance skin flap survival [3]. Surgical delay involves partially interrupting the blood supply to the flap while preserving partial circulation (e.g., through veins or small branch vessels), thereby promoting angiogenesis in ischemic tissue and improving the flap’s blood supply, stability, and survival rate [40]. IPC, on the other hand, uses short, repeated cycles of occlusion and reperfusion to help tissues adapt to the loss of blood supply. This non-invasive strategy has been shown to effectively reduce ischemia-reperfusion (I/R) injury in flaps by promoting angiogenesis, thereby improving blood flow and further enhancing flap survival [41]. The effectiveness varies due to differences in vascular responses among patients and requires proper timing and strict control; otherwise, it may damage the flap. Its mechanism is not yet fully understood, and clinical application is influenced by factors such as patient age, medications, and underlying conditions [42]. It relies on advanced equipment and close monitoring, with discrepancies being observed between experimental and clinical outcomes. Hyperbaric oxygen therapy enhances skin flap survival by increasing tissue oxygen supply and promoting angiogenesis. However, it requires specialized equipment, multiple sessions, and may lead to complications such as oxygen toxicity, especially in areas with limited resources [43]. To overcome limitations, researchers are exploring the applicability of approaches including stem cell-based and biomaterial-based methods. Adipose-derived stem cells (ADSCs) improve angiogenesis in skin flaps by regulating the hypoxia-inducible factor 1-alpha (HIF-1α)/VEGF pathway [44]. Additionally, a meta-analysis showed that stem cell therapy significantly increased blood vessel density and enhanced the expression of VEGF [38]. Subgroup analysis revealed that the effectiveness of stem cell therapy in improving skin flap survival varied based on factors such as the skin flap area, cell type, transplantation method, and stem cell administration approach [38]. Subcutaneous injection of exogenous VEGF or VEGF viral vectors can enhance skin vitality in skin flap surgery, with the mechanism involving increased angiogenesis and the synthesis/release of the vasodilator factor NO [45,46]. A decellularized skin/adipose tissue flap (DSAF) repopulated with human adipose-derived stem cells (hADSCs) and human umbilical vein endothelial cells (HUVECs) promotes neovascularization and tissue remodeling, providing a promising platform for vascularized soft-tissue engineering following anastomosis in nude mice [47]. HUVECs were seeded onto pre-vascularized human mesenchymal stem cell (hMSC) sheets and applied in a skin flap animal model, significantly improving angiogenesis and microcirculation [48]. Despite numerous studies on stem cell therapy in animal skin flap models, clinical trials investigating the use of stem cells to enhance flap survival have not yet been conducted. Moreover, research suggests that employing stem cells for free flap reconstruction following mastectomy or cancer resection may increase the risk of tumor proliferation and metastasis. Therefore, further studies are needed to clarify the potential benefits of various stem cell applications in reconstructive surgery [3].

## 3. Exosomes: Biogenesis, Composition, and Function

### 3.1. Biogenesis of Exosomes: Origin and Release Mechanisms

As shown in Figure 2, exosome biogenesis is closely linked to the endocytic pathway, beginning with the invagination of the plasma membrane to form early exosomes (EEs). These EEs mature into late endosomes (LEs) and subsequently into multivesicular bodies (MVBs) containing intraluminal vesicles (ILVs) [49]. MVBs can either fuse with lysosomes for degradation or with the plasma membrane to release ILVs as exosomes into the extracellular space [50,51]. The secretion of exosomes into the extracellular space is facilitated by proteins from the ras-related in brain GTPase (Rab-GTPase) family, such as RAB2B, RAB5A, RAB7, RAB9A, RAB11, RAB27, and RAB35. Additionally, soluble N-ethylmaleimide-sensitive factor Attachment Receptor (SNARE) family proteins, including vesicle-associated membrane protein 7 (VAMP7) and Ykt6 t-SNARE protein (YKT6), have also been associated with this release process [52,53,54]. Exosome biogenesis occurs through three primary pathways: first, vesicles bud into discrete endosomes that develop into MVBs, which then release exosomes upon fusion with the plasma membrane; second, vesicles can directly bud from the plasma membrane; and third, a delayed release mechanism allows budding at intracellular plasma membrane-connected compartments (IPMCs), followed by the deconstruction of IPMC necks [55]. The transport of MVBs relies on various accessory proteins, including TSG101 (tumor susceptibility gene 101 protein), ALIX (apoptosis-linked gene 2-interacting protein X, encoded by PDCD6IP), VTA1 (vesicle trafficking), and VPS4 (vacuolar protein sorting-associated protein 4) [56,57,58,59,60], which play crucial roles in both the endosomal sorting complexes required for transport (ESCRT)-dependent and ESCRT-independent exosome formation.

The ESCRT pathway, the primary mechanism for sorting ubiquitinated proteins into ILVs, consists of four complexes (ESCRT-0, -I, -II, and -III) that cooperate to facilitate cargo sorting and vesicle budding. ESCRT-0 is activated by phosphatidylinositol 3-phosphate (PI(3)P) and identifies ubiquitinated transmembrane proteins on the endosomal membrane, promoting their concentration [56]. It recruits ESCRT-I, which is crucial for sorting cargo and deforming the membrane to form buds. ESCRT-II also aids in cargo sorting and regulates the formation of ESCRT-III [61,62]. Finally, ESCRT-III is responsible for sorting MVB cargo, driving vesicle scission, and facilitating the recycling of ESCRT components via vacuolar protein sorting 4 (VPS4) ATPase [63]. The ESCRT-independent pathways for exosome formation highlight alternative mechanisms beyond the well-established ESCRT pathway. Key points include the role of heparan sulfate proteoglycans and the cytosolic protein syntenin, which facilitate the interaction between syndecan and ESCRT proteins [64]. This process aids in the budding and scission of MVBs and ILVs. Notably, exosome secretion can occur in oligodendroglial cells without ESCRT involvement, emphasizing ceramide’s crucial role in sorting [65]. Additionally, tetraspanin proteins and other molecules, such as flotillin-2 and phospholipase D2, contribute to cargo selection and exosome formation [66,67]. Overall, the complexity of exosome biogenesis suggests multiple pathways and mechanisms.

### 3.2. Composition of Exosomes: Proteins, Lipids, and RNA

Exosomes consist of a wide range of substances, as reported in the latest update of the exosome database (http://www.exocarta.org, accessed on 31 January 2025), which includes 13,472 proteins, 3408 messenger RNA (mRNAs), 10755 microRNAs (miRNAs), and 3946 lipids. These components can function as autocrine and/or paracrine factors, encompassing specific lipids, proteins, DNA, mRNAs, and noncoding RNAs.

Exosomes are intricate extracellular vesicles that encapsulate a broad spectrum of proteins, and which play pivotal roles in numerous physiological and pathological processes. Among these proteins, ubiquitous proteins, such as cytoskeletal elements like tubulin and actin, as well as annexins and Rab proteins, are prominent. These proteins are crucial for facilitating intracellular membrane fusion and transport, thereby contributing to exosome biogenesis and functionality [68]. Heat-shock proteins (HSPs), particularly HSP70 and HSP90 [69,70], are significantly represented in exosomes and serve essential functions in protein folding, ensuring that other proteins maintain their proper conformation and function. They also play a crucial role in antigen presentation, as they can bind antigenic peptides and assist in loading them onto major histocompatibility complex (MHC) molecules. The presence of MHC class I and class II molecules in exosomes is indicative of their role in immune modulation, making them important players in antigen presentation and immune response [71]. Additionally, tetraspanins, which include CD9, CD63, CD81, and CD82, are highly enriched in exosomes and are known to facilitate the organization of molecular complexes and membrane subdomains. These tetraspanins interact with various protein partners, including integrins and MHC molecules, which underscores their role in cellular signaling and communication [72]. Exosomes also harbor cell-specific proteins that reflect the functions and identity of the originating cells. For example, antigen-presenting cells (APCs) contain abundant MHC class II molecules and CD86, a crucial co-stimulatory molecule for T-cell activation. T-cell-derived exosomes are enriched with T-cell receptors, further highlighting the role of exosomes in immune responses. Additionally, exosomes from different cell types contain specific transmembrane proteins, such as integrins and immunoglobulin-family members, which facilitate targeted communication with recipient cells [73]. Furthermore, recent proteomic studies have identified a subset of proteins that are selectively targeted to exosomes, revealing conservation of protein composition across species, with about 80% of proteins in exosomes from mouse and human dendritic cells (DCs) being conserved [74,75]. This suggests a fundamental mechanism governing exosome biogenesis and function. Importantly, the presence of these proteins in exosomes highlights their potential as biomarkers for various diseases and their utility in intercellular communication, tumor progression, and immune regulation.

The lipid composition of exosomes is still not well-characterized, with limited data available. Generally, exosomes derived from reticulocytes (or leukocytes) exhibit a lipid composition similar to that of the plasma membrane of the parent cell [76]. For instance, B-cell-derived exosomes have been reported to contain lyso-bis-phosphatidic acid, a lipid that is predominantly found in late endocytic compartments [77]. Phosphatidylserine (PS), typically located on the cytosolic side of the plasma membrane, is also detected in exosomes derived from platelets and DCs, although at lower concentrations [78]. Additionally, cholesterol is abundant in the internal vesicles of late endosomes and in exosomes from Epstein–Barr virus (EBV)-transformed B cells, mirroring the composition of lipid rafts found in plasma membrane microdomains [79]. The lipid profiles of exosomes from other cell types remain largely unexplored. Collectively, these studies underscore that exosomes represent a distinct class of secreted sub-cellular compartments.

Furthermore, exosomes serve as notable carriers of various types of RNA, including miRNAs, long noncoding RNAs (lncRNAs), and circular RNAs (circRNAs). miRNAs are small noncoding RNAs that regulate gene expression by targeting mRNAs, and their presence in exosomes has been studied extensively as potential biomarkers for disease prognosis and progression [80,81]. Similarly, lncRNAs, which are transcripts longer than 200 nucleotides, are selectively packaged into exosomes and participate in cell communication, influencing processes like growth, differentiation, and tissue remodeling [82]. Exosomal circRNAs, a new family of noncoding RNAs with tissue-specific expression patterns, regulate gene expression by acting as miRNA sponges, contributing to multiple cellular functions [83].

### 3.3. General Functions of Exosomes in Intercellular Communication

Exosomes play a crucial role in intercellular communication by transmitting various bioactive molecules, including proteins, lipids, and nucleic acids, which influence the functions of recipient cells. These small vesicles utilize multiple mechanisms to mediate the exchange of information. Firstly, exosomes can directly stimulate target cells via surface-bound ligands or transfer activated receptors to the recipient cell membrane, altering signal transduction [84,85]. Additionally, exosomes are capable of transferring genetic materials, such as mRNA and miRNA, to reprogram the epigenetic landscape of recipient cells, thereby regulating physiological or pathological processes. In terms of immune regulation, exosomes participate in immune response modulation through antigen presentation, immune activation, and immune suppression, by such means as promoting antigen presentation, inducing immune tolerance, and suppressing inflammation. MSC-derived exosomes also play a significant role in wound healing, promoting tissue repair, inhibiting inflammation, and accelerating cell migration and regeneration [86]. In summary, exosomes not only serve as essential mediators of intercellular communication but also regulate various physiological and pathological processes, including angiogenesis, improvement of cardiac function, modulation of immune responses, anti-fibrosis, and anti-apoptosis.

## 4. Exosomes and Angiogenesis

### 4.1. Exosomal Cargo and Angiogenesis

Exosomes promote angiogenesis by delivering various bioactive molecules to target cells, including miRNAs, lncRNAs, circRNAs, and proteins. These molecules modulate critical processes essential for new blood vessel formation, such as endothelial cell proliferation, migration, and tube formation.

#### 4.1.1. miRNAs

Exosomes derived from MSCs have garnered increasing attention for their ability to influence various biological processes, including blood vessel formation and tissue repair [87]. These exosomes carry miRNAs that can regulate endothelial cell behavior and impact vascular development [88,89]. Below are several examples of how MSC-derived exosomal miRNAs contribute to vascular processes. Exosomes from MSCs containing the miR-23a family promote endothelial cell migration and blood vessel formation by targeting Sema6A and Sprouty2. These miRNAs are notably overexpressed in lung and heart tissues [87]. Exosomes from human umbilical cord mesenchymal stem cells (hucMSCs) carry miR-126-3p, which targets the phosphoinositide-3-kinase regulatory subunit 2 (PIK3R2) gene and activates the phosphoinositide 3-kinase/protein kinase B/mechanistic target of rapamycin (PI3K/AKT/mTOR) pathway, enhancing endothelial cell proliferation and supporting blood vessel growth while inhibiting apoptosis, ultimately improving ovarian angiogenesis and functional recovery [90]. Exosomal miR-424-5p from hypoxic bone marrow MSCs enhances vessel formation by targeting Delta-like canonical Notch ligand 4 (DLL4), which in turn activates the DLL4/Notch signaling pathway and increases pro-angiogenic gene expression, aiding in endometrial injury repair [91]. MicroRNA-132 delivered via exosomes from MSCs downregulates the expression of the negative regulator Ras p21 protein activator 1 (RASA1) in ECs, thereby supporting vessel formation in myocardial infarction [92]. Beyond their regenerative and repair functions, MSC-derived exosomes also serve as key players in the tumor microenvironment, particularly in vascular remodeling and tumor progression. For instance, miR-16 inhibits blood vessel formation by downregulating VEGF expression in breast cancer cells [93]. MicroRNA-100 delivered by MSC exosomes reduces in vitro vessel growth by regulating the mTOR/HIF-1α/VEGF signaling pathway in breast cancer cells [94]. Exosomal miR-133b-3p from bone marrow MSCs suppresses vessel formation and oxidative stress by repressing FBN1 in diabetic retinopathy [95].

miRNAs play a crucial role in regulating the partial endothelial-to-mesenchymal transition (EndoMT) during neovascularization [96]. Exosomes derived from high glucose-treated ARPE19 cells transfer miR-202-5p to HUVECs, suppressing EndoMT by targeting transforming growth factor beta receptor 2 (TGFβR2) and modulating the TGF/SMAD pathway [97]. Exosomal miR-21-3p promotes angiogenesis by inhibiting PTEN and sprouty homolog 1 (SPRY1), thereby enhancing endothelial cell function [98].

miR-214 silences ataxia telangiectasia mutated, preventing senescence and supporting blood vessel formation [99]. Exosomal miRNAs are involved in communication both within homogeneous ECs and in signaling between different types of cells. Exosomal miR-92a secreted by leukemia cells (K562) transfers into HUVECs and inhibits integrin α5 expression, promoting endothelial cell migration and tube formation during tumor angiogenesis [100].

Several studies highlight how exosomal miRNAs derived from different cancer cells contribute to the regulation of endothelial cell behavior, promoting vascular development, permeability, and metastasis. Below are examples of how exosomal miRNAs influence angiogenesis and related processes in different cancer types. Exosomal miR-92a-3p derived from retinoblastoma cells promotes angiogenesis by increasing endothelial cell migration and tube formation through downregulation of Krüppel-like factor 2 (KLF2), a key regulator of angiogenesis [101]. Exosomal miR-21-5p secreted by colorectal cancer cells activates the β-catenin signaling pathway in ECs by targeting Krev interaction trapped 1 (KRIT1), which enhances angiogenesis and vascular permeability in colorectal cancer (CRC) [102]. Exosomal miR-3157-3p from non-small cell lung carcinoma (NSCLC) cells promotes pre-metastatic niche formation, angiogenesis, and increased vascular permeability by targeting tissue inhibitor of metalloproteinase (TIMP)/KLF2, which regulates the expression of VEGF, MMP2, MMP9, and occludin in ECs [103]. Exosomal miR-320d from colorectal cancer cells promotes tumor spread and vascular growth by targeting G protein subunit alpha i1 (GNAI1) in ECs, triggering janus kinase 2 (JAK2)/signal transducer and activator of transcription 3 (STAT3) activation and vascular endothelial growth factor A (VEGFA) production [104]. Exosomal miR-210 secreted by hepatocellular carcinoma (HCC) cells is transferred into ECs, where it directly downregulates the expression of Sma and Mad-related protein 4 (SMAD4) and signal transducer and activator of transcription 6 (STAT6), thereby promoting angiogenesis and driving tumor vasculature formation [105]. Exosomal miR-23a from hypoxic lung cancer cells promotes blood vessel development and increases vascular permeability by suppressing prolyl hydroxylase domain-containing 1 and 2 (PHD1/2) and zonula occludens-1 (ZO-1), thus enhancing tumor vasculature and facilitating transendothelial migration [106]. M2 macrophage-derived exosomal miR-501-3p promotes the progression of pancreatic ductal adenocarcinoma (PDAC) by downregulating the tumor suppressor gene TGFBR3 and activating the TGF-β signaling pathway [107].

#### 4.1.2. LncRNAs

Tumor-derived exosomes lncRNAs are involved in tumor angiogenesis [108,109]. LncRNA linc-CCAT2, transferred via glioma-derived exosomes, promotes angiogenesis by upregulating VEGFA and TGFβ, enhancing endothelial cell proliferation, migration, and survival while inhibiting apoptosis [110]. Exosomal lncRNA OIP5-AS1 promotes osteosarcoma progression by regulating angiogenesis and autophagy via the miR-153/autophagy-related 5 (ATG5 axis) [111]. CD90+ liver cancer cells promote an angiogenic phenotype in ECs through exosomal release of lncRNA H19 [112]. Exosomal lncRNAs can act as competitive endogenous RNAs (ceRNAs) by binding to miRNAs, inhibiting their regulation of target genes. This “sponging” effect prevents miRNAs from targeting their genes, thereby activating angiogenesis-related signaling pathways [113]. Exosomal lncRNA H19 promotes osteogenesis and angiogenesis by acting as a “sponge” for miR-106a, which upregulates Angpt1 expression and activates the lnc-H19/Tie 2 receptor- Nitric Oxide (Tie2-NO) signaling pathway in mesenchymal and ECs [114]. Exosomal lncRNA UCA1, derived from hypoxic pancreatic tumors, facilitates angiogenesis by regulating the miR-96-5p/Angiomotin-like protein 2 (AMOTL2) pathway [115].

#### 4.1.3. CircRNAs

Exosomal circRNA-100,338 regulates angiogenesis by interacting with the RNA-binding protein neuro-oncological ventral antigen 2 (NOVA2), activating the mTOR pathway, thereby promoting hepatocellular carcinoma metastasis [116]. Similarly, exosomal circRNAs can act as miRNA sponges to regulate angiogenesis. Exosomal circ29 enhances angiogenesis in gastric cancer by sequestering miR-29a, leading to the activation of the VEGF pathway [117]. Exosomal circSHKBP1 sponges miR-582-3p to upregulate human antigen R (HUR) expression, enhancing VEGF mRNA stability and promoting tumor angiogenesis [118]. cPWWP2A, acting as a miR-579 sponge, regulates pericyte biology and endothelial cell crosstalk via exosomes and affects angiogenesis through the angiopoietin 1/occludin/Sirtuin 1 (SIRT1) network [119]. The delivery of circ-Snhg11 through hypoxia-pretreated adipose-derived stem-cell-derived exosomes (ADSC-exos) embedded in gelatin methacryloyl (GelMA) hydrogels enhances endothelial cell function and promotes diabetic wound healing by activating the miR-144-3p/Nuclear factor, erythroid 2 like 2 (NFE2L2)/hypoxia-inducible factor 1-alpha (HIF1α) signaling pathway [120].

#### 4.1.4. Proteins

Exosomes derived from hypoxic stem cells of the apical papilla (SCAPs) promote angiogenesis in HUVECs by activating the HIF-1α/Notch1 signaling pathway, which enhances the delivery of jagged-1 (JAG1) and subsequently increases VEGF production [121]. Annexin II in exosomes (exo-Anx II) enhances blood vessel formation by promoting tissue plasminogen activator (tPA)-dependent angiogenesis [122]. Exosome-associated DLL4 promotes angiogenesis by transferring to ECs, inhibiting Notch signaling, and inducing a tip cell phenotype in ECs [123]. Tetraspanin Tspan8-containing exosomes promote angiogenesis by enhancing endothelial cell proliferation, migration, and maturation through VEGF-independent regulation of angiogenesis-related genes [124]. Exosomes carrying latent TGF-β and associating with the transmembrane proteoglycan betaglycan can activate SMAD-dependent signaling pathways and significantly increase the production of fibroblast growth factor 2 (FGF2) in fibroblasts [125]. Exosomes enriched with Src, insulin-like growth factor 1 receptor (IGF-IR), and focal adhesion kinase (FAK) proteins activate Src/FAK signaling, enhancing prostate cancer cell proliferation, migration, and angiogenesis, which may provide potential biomarkers for prostate cancer (PrCa) progression [126]. Exosomes from high-grade ovarian cancer promote angiogenesis by carrying proteins like activating transcription actor 2 (ATF2) and metastasis-associated protein 1 (MTA1), which activate VEGF and HIF-1 pathways, boosting tumor growth and spread [127]. Exosomes derived from the urine of patients with high-grade bladder cancer contain bioactive molecules such as EGF-like repeats and discoidin I-like domain 3 (EDIL-3), which can promote angiogenesis and bladder cancer cell migration by activating the epidermal growth factor receptor (EGFR) signaling pathway [128].

A list of all proteins and noncoding RNAs presented in exosomes derived from various sources and involved in regulating angiogenesis, according to the studies mentioned, is provided in Table 1, Table 2, Table 3 and Table 4, highlighting the multi-layered regulatory mechanisms and underscoring the crucial role of exosomes in promoting angiogenesis.

This table summarizes the roles of exosomal microRNAs (miRNAs) derived from mesenchymal stem cells (MSCs) and tumor cells in regulating angiogenesis. It highlights how MSC-derived exosomal miRNAs, such as miR-23a, miR-126-3p, miR-424-5p, and miR-132, promote blood vessel formation and tissue repair through various mechanisms, including the activation of key signaling pathways like PI3K/AKT/mTOR and DLL4/Notch. Additionally, the table describes how tumor-derived exosomal miRNAs, such as miR-92a, miR-21-5p, and miR-320d, contribute to tumor angiogenesis, metastasis, and vascular permeability by modulating endothelial cell behavior and related signaling pathways. Abbreviations: miRNAs (MicroRNAs); MSC (Mesenchymal Stem Cell); Sema6A (Semaphorin 6A); Sprouty2 (Sprouty homolog 2); hucMSC (human umbilical cord mesenchymal stem cells); PIK3R2 (phosphoinositide-3-kinase regulatory subunit 2); PI3K (Phosphoinositide 3-Kinase); AKT (Protein Kinase B); mTOR (mechanistic Target of Rapamycin); DLL4 (Delta-like canonical Notch ligand 4); RASA1 (Ras p21 protein activator 1); VEGF (Vascular Endothelial Growth Factor); HIF-1α (Hypoxia-Inducible Factor 1-alpha); FBN1 (Fibrillin 1); ARPE19 (Adult Retinal Pigment Epithelial cell line 19); EndoMT (Endothelial-to-Mesenchymal Transition); TGFβR2 (Transforming Growth Factor Beta Receptor 2); TGF (Transforming Growth Factor); Smad (Sma and Mad-related protein); PTEN (Phosphatase and Tensin Homolog); SPRY1(Sprouty RTK Signaling Antagonist 1); Integrin α5 (Integrin alpha5); KLF2 (Krüppel-Like Factor 2); KRIT1 (Krev interaction trapped 1); TIMP (Tissue Inhibitor of Metalloproteinase); MMPs (Matrix Metalloproteinase); GNAI1 (G protein subunit alpha i1); JAK2 (Janus kinase 2); STAT3 (Signal Transducer and Activator of Transcription 3); SMAD4 (SMAD Family Member 4); STAT6 (Signal Transducer and Activator of Transcription 6); PHD1/2 (Prolyl Hydroxylase Domain-containing 1 and 2); ZO-1 (Zonula Occludens-1); TGFBR3 (Transforming Growth Factor Beta Receptor 3); TGF-β (Transforming Growth Factor Beta).

LncRNAs play a critical role in tumor angiogenesis by influencing endothelial cell. behavior and modulating angiogenesis-related pathways. LncRNAs such as linc-CCAT2, OIP5-AS1, H19, and UCA1 regulate the expression of key factors like VEGFA, TGFβ, and Angpt1, as well as control signaling pathways like miR-153/ATG5 and miR-96-5p/AMOTL2. Some lncRNAs, such as H19, act as competitive endogenous RNAs (ceRNAs), “sponging” miRNAs and preventing them from targeting their respective genes, thereby activating angiogenesis and promoting tumor progression. Abbreviations: lncRNAs (Long non coding RNAs); linc-CCAT2 (long intergenic non-coding RNA colon cancer-associated transcript 2); VEGFA (Vascular Endothelial Growth Factor A); TGFβ (Transforming Growth Factor Beta); OIP5-AS1(Opa-interacting protein 5 antisense RNA 1); ATG5 (Autophagy-related 5); VEGF (Vascular Endothelial Growth Factor); HUVEC (Human Umbilical Vein Endothelial Cells); Angpt1 (Angiopoietin-1); Tie2 (Tie 2 receptor); NO (Nitric Oxide); AMOTL2 (Angiomotin-like protein 2).

A summary of exosomal circular RNAs (circRNAs) in angiogenesis, focusing on the roles of circR NAs like circRNA-100,338, circ29, circSHKBP1, and circ-Snhg11 in regulating endothelial cell behavior through interactions with RNA-binding proteins, miRNA sponging, and modulation of signaling pathways such as mTOR, VEGF, and angiopoietin. Abbreviations: circRNAs (Circular RNAs); NOVA2 (Neuro-oncological ventral antigen 2); mTOR (Mechanistic target of rapamycin); VEGF (Vascular Endothelial Growth Factor); HUR (Human antigen R); SIRT1 (Sirtuin 1); ADSCs (Adipose-derived stem cells); NFE2L2 (Nuclear factor, erythroid 2 like 2); HIF1α (Hypoxia-inducible factor 1-alpha).

A summary of the role of exosomal proteins in angiogenesis, highlighting how exosomes derived from various sources, such as hypoxic stem cells, prostate cancer cells, and high-grade ovarian cancer cells, promote angiogenesis. Key proteins, including jagged-1 (JAG1), Annexin II, Dll4, Tetraspanin Tspan8, latent TGF-β, and bioactive molecules like EDIL-3, enhance endothelial cell proliferation, migration, and vascular development through signaling pathways such as HIF-1α/Notch1, Src/FAK, and EGFR, contributing to tumor progression and tissue repair. Abbreviations: SCAPS (Stem Cells of the Apical Papilla); HIF-1α (Hypoxia-Inducible Factor 1 Alpha); JAG1 (Jagged-1); HUVEC (Human Umbilical Vein Endothelial Cells); Notch1 (Notch Receptor 1); VEGF (Vascular Endothelial Growth Factor); Annexin II (Annexin A2); tPA (Tissue Plasminogen Activator); Dll4 (Delta-like Ligand 4); Tspan8 (Tetraspanin 8); TGF-β (Transforming Growth Factor Beta); SMAD (Sma and Mad); FGF2 (Fibroblast Growth Factor 2); IGF-IR (Insulin-like Growth Factor 1 Receptor); FAK (Focal Adhesion Kinase); ATF2 (Activating Transcription Factor 2); MTA1 (Metastasis-associated Protein 1); HIF-1 (Hypoxia-Inducible Factor 1); EDIL-3 (EGF-like repeats and discoidin I-like domain 3); EGFR (Epidermal Growth Factor Receptor).

### 4.2. Comparison with Other Pro-Angiogenic Factors

Compared to conventional pro-angiogenic factors such as VEGF, exosomes offer a more nuanced and multifaceted approach to promoting angiogenesis. While traditional factors primarily stimulate endothelial cell proliferation and migration through singular pathways, exosomes carry a complex cargo of miRNAs, proteins, and lipids that can simultaneously target multiple signaling cascades. Furthermore, exosomes facilitate a more sustained and coordinated cellular response, amplifying the effects of other angiogenic signals over time. Unlike conventional factors that may exert transient effects, exosomes possess the ability to release signaling molecules persistently, ensuring ongoing stimulation of angiogenesis even under hypoxic conditions. This multi-layered regulatory capability positions exosomes as a superior or complementary option to traditional pro-angiogenic factors in therapeutic applications.

## 5. Therapeutic Potential of Exosomes in Enhancing Skin Flap Survival

### 5.1. Exosome-Based Therapies (Figure 3)

Exosomes, especially those derived from stem cells, have emerged as powerful therapeutic agents for enhancing skin flap survival. Macrophage-derived exosomes (M2-exosomes) enhance the survival area and microvascular density of skin flaps by upregulating HIF-1α and VEGFA and downregulating hypoxia-inducible factor 1-alpha inhibitor (HIF1AN), promoting the proliferation, migration, and tube formation of HUVECs [129]. Exosomes derived from HUVECs under oxidative stress significantly enhanced the pro-angiogenic ability of endothelial progenitor cells (EPCs) through the Wnt/β-catenin signaling pathway mediated by Lnc NEAT1, thereby improving random skin flap survival [130]. Hypoxic preconditioned bone marrow stromal cell (BMSC)-derived exosomes (Hypo-Exo) carry miR-421-3p to target and regulate the mTOR pathway, increasing the phosphorylation levels of Unc-51, like autophagy activating kinase 1 (ULK1) and FUN14 domain containing 1 (FUNDC1), and thereby activating autophagy and promoting flap survival after ischemia-reperfusion injury [131]. BMSC-derived exosomes promote flap survival and mitigate ischemia-reperfusion injury in a free abdominal-flap rat model by reducing oxidative stress, inflammation, and apoptosis, while enhancing angiogenesis [132]. Using a self-healing oxidized pullulan-carboxymethylated chitosan composite hydrogel as a carrier, curcumin-loaded exosomes improve flap survival rates [133]. Sequential transplantation of exosomes and hypoxia-pretreatment significantly improved the survival rate of the trans-territory perforator flap necrosis model [134]. H_2_O_2_-ADSC-exos preconditioned with low concentrations of H_2_O_2_ promotes skin flap survival after ischemia-reperfusion injury by enhancing neovascularization, reducing inflammation, and inhibiting apoptosis [135]. Exosomes derived from ADSCs enhance neovascularization and blood supply in skin flaps, thereby improving their survival rate following transplantation [136]. Human dental pulp stem cell (hDPSC)-exos enhance vein endothelial cell growth, movement, and tube formation in a dose-dependent manner through the activation of the PI3K/AKT pathway, significantly improving flap survival rates and microvessel density while reducing epithelial cell apoptosis in a rat model [137]. ADSC-exos reduce ischemia/reperfusion injury through the secretion of IL-6, which stimulates angiogenesis and supports the differentiation of ADSCs into ECs [138]. Bioinformatics analysis indicated that hsa-miR-760 is significantly upregulated while hsa-miR-423-3p is significantly downregulated in ADSC-exos, and these miRNAs may regulate the expression of the integrin subunit alpha 5 (ITGA5) and histone deacetylase 5 (HDAC5) genes, respectively, to enhance the vascularization of skin flaps [139]. Exosomes derived from BMSCs improve skin flap viability in rats by enhancing angiogenesis and increasing microvascular density through elevated expression of VEGF and cluster of differentiation 34 (CD34) [140].

**Figure 3 biomedicines-13-00353-f003:**
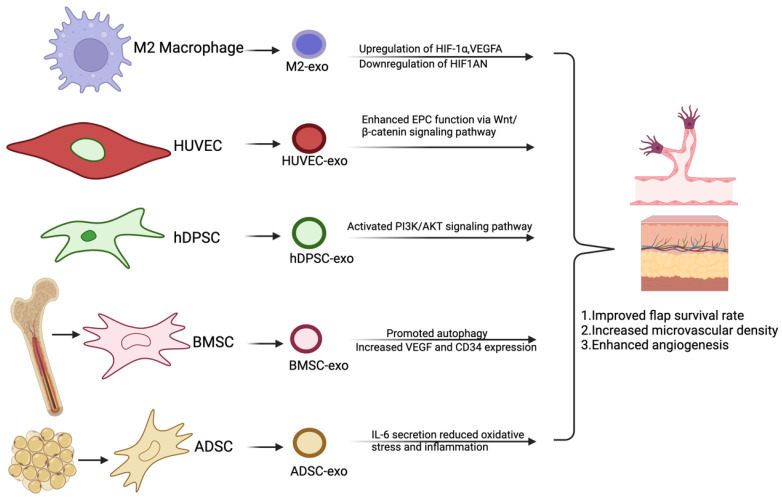
Exosomes derived from various stem cells, such as M2 macrophages, HUVECs, BMSCs, ADSCs, and hDPSCs, enhance skin flap survival and vascularization. These exosomes promote angiogenesis, reduce ischemia-reperfusion injury, and improve flap survival by upregulating key factors like HIF-1α, VEGFA, miR-421-3p, and VEGF, as well as activating signaling pathways such as Wnt/β-catenin, PI3K/AKT, and mTOR. Additionally, they stimulate endothelial cell proliferation, migration, and tube formation, while mitigating oxidative stress, inflammation, and apoptosis, highlighting the clinical potential of exosome-based therapies for tissue regeneration and repair. Exosome-based therapeutics for enhancing skin flap survival and vascularization. Abbreviations: HUVEC (human umbilical vein endothelial cells); hDPSC (human dental pulp stem cells); BMSC (bone marrow stromal cell); ADSC (adipose-derived stem cells); HIF-1α (hypoxia-inducible factor 1-alpha); VEGFA (vascular endothelial growth factor A); HIF1AN (hypoxia-inducible factor 1-alpha inhibitor); EPCs (endothelial progenitor cells); PI3K (phosphoinositide 3-kinase); AKT (protein kinase B); VEGF (vascular endothelial growth factor); CD34 (cluster of differentiation 34); and IL-6 (interleukin 6).

In summary, these studies indicate that exosomes have significant therapeutic potential in skin flap survival, especially in enhancing angiogenesis, reducing ischemic damage, and promoting tissue regeneration. Exosomes derived from different sources of stem cells exert their effects through various mechanisms, highlighting their important clinical application potential. As minimally invasive and effective therapeutic agents, exosomes offer exciting prospects for improving skin flap survival and tissue regeneration. Nevertheless, most studies still rely on animal models, particularly rat models, to evaluate the effects of exosomes on skin flap survival. Although these models provide preliminary evidence, the differences between animal and human physiological conditions may lead to inconsistent results in clinical applications. Therefore, more clinical trials and human sample data are needed to validate the therapeutic potential of exosomes. The studies mention exosomes from different sources, such as M2 macrophages, HUVECs, BMSCs, and ADSCs. However, exosomes from various sources may differ in function, composition, and mechanisms of action. The lack of standardized protocols and comparisons may reduce the comparability of different study results, which, in turn, could affect the feasibility of clinical translation. Most current studies mainly focus on the short-term effects of exosomes on skin flap survival, but there is limited evaluation of long-term efficacy and tissue regeneration. The long-term outcomes of skin flap transplantation and the potential side effects of exosome therapy, such as excessive angiogenesis or immune responses, need further attention.

### 5.2. Challenges in Exosome Therapy

Exosomes have demonstrated the ability to repair adverse conditions and damaged tissues. To enhance the therapeutic potential of exosomes, several strategies have been employed in laboratories, such as preconditioning stem cells, genetically modifying them, and combining exosomes with biomaterials to boost their production and efficacy [141,142,143]. However, there are still many issues and challenges to be addressed in its practical clinical application.

The purity and physicochemical properties of exosomes are primarily influenced by the separation methods used. Collecting high-quality and homogeneous exosomes is a challenge that needs to be addressed. The complexity of exosome isolation arises from the diversity of biological samples, the overlap in physicochemical and biochemical properties between exosomes and other EVs, as well as the inherent heterogeneity of exosomes themselves [144]. Ensuring quality consistency across different exosome preparations is crucial for reliable therapeutic applications. Various techniques, including ultracentrifugation, ultrafiltration, size exclusion chromatography (SEC), immunoaffinity capture, liquid-liquid phase systems (LLPS), microfluidics, and polymer-based precipitation, can be applied for large-scale exosome production [144,145]. For instance, differential centrifugation is a commonly used method for exosome isolation but may lead to contamination with non-exosomal impurities, making it unsuitable for clinical applications [146]. Ultrafiltration is prone to membrane clogging and vesicle capture issues, leading to reduced efficiency [147]. SEC can isolate high-purity exosomes but requires long run times, limiting its high-throughput applications [148]. Immunoaffinity capture allows for selective isolation of exosomes from specific sources, yielding high purity, but the yield is low, and antigen regulation issues may lead to erroneous results, thus affecting quality consistency [149]. Exosome precipitation is simple to perform but yields are unstable and prone to co-precipitation with other cellular components, affecting purity and subsequent analysis [150]. Different isolation techniques can lead to variations in the concentration, purity, size, profile, and contents of exosomes, further impacting quality consistency [151,152,153]. For example, the mRNA expression patterns may differ in subsequent RNA sequencing analysis [154,155]. Common co-isolated proteins, such as albumin, immunoglobulins, and MMPs, are abundant in body fluids, and these “contaminants” may affect the purity of exosomes [156]. Achieving absolute purity is challenging. To improve separation efficiency, researchers have attempted to combine different techniques, which can, in some cases, enhance the separation outcomes, but this also increases costs and operational complexity. Furthermore, the source cells of exosomes are highly heterogeneous, making the separation and identification of different subpopulations a continuing challenge [157]. Although most separation techniques are used in basic research, throughput and validation for clinical applications still need to be addressed. Therefore, exosome separation and detection technologies must undergo large-scale evaluation using clinical samples to ensure their sensitivity and selectivity, quality consistency, and to validate their clinical applicability. Moreover, to meet the demands for low cost, reliability, and speed in clinical diagnosis and treatment, existing separation techniques need further improvement to enhance their scalability and efficiency, especially when processing large volumes of human blood and plasma samples. By optimizing these technologies and combining them with appropriate screening systems, the application of exosomes as drug carriers and biomarkers can be improved, ensuring drug delivery efficiency and diagnostic accuracy. Furthermore, protecting the contents during the separation process also impacts therapeutic outcomes, helping to prevent RNA and protein degradation. Subtype separation of exosomes can enhance treatment targeting, and advances in standardized and high-throughput separation techniques will contribute to improving the quality consistency, reliability, and efficiency of clinical treatments. This is also the foundation for the standardization of exosome therapeutic dosages.

Some studies have raised concerns about the potential side effects, safety, and adverse outcomes of exosome therapy, highlighting the need for extensive clinical trials to further evaluate these issues [158,159]. For example, exosome applications, especially those involving xenogeneic exosomes, may provoke immune system responses [160]. The stability and metabolic pathways of exosomes in the body are also critical safety considerations. Questions about whether exosomes can be effectively cleared from the body, whether they may accumulate, or whether they might cause toxicity in organs such as the liver and spleen need to be addressed through long-term monitoring and assessment [161]. Although exosomes have been widely studied for cancer therapy, certain tumor-derived exosomes may contain factors that promote tumor growth or metastasis [162]. If not controlled, these exosomes could inadvertently accelerate cancer progression. Therefore, it is essential to closely monitor the source and biological activity of exosomes to ensure therapeutic efficacy rather than adverse effects.

Pursuing high purity separation methods often leads to low yield, which may limit the feasibility of exosome use in large-scale clinical applications. Obtaining a sufficient quantity of exosomes for clinical use remains a significant challenge. The yield of exosomal protein is generally less than 1 µg per mL of culture medium, while most studies suggest that an effective dose of exosomes is around 10–100 µg exosomal protein per mouse [163,164,165,166]. Additionally, the storage conditions need to be standardized [167]. To meet clinical demands, it is necessary to optimize the exosome production process. Future research should focus on improving the efficiency and scale of exosome production, especially for exosomes derived from multiple cell types. It is anticipated that bioreactors, large-scale cell culture, and automation technologies could be employed to significantly increase the yield.

Future research needs to place greater emphasis on collaboration between preclinical and clinical stages to promote clinical trial studies of exosome-based therapies. As of now, there are a total of 104 registered clinical trials on exosome therapy (from ClinicalTrials.gov). Through the design of well-structured clinical trials, the safety and efficacy of exosomes in treating various diseases will be validated, ultimately advancing exosome therapy from the laboratory to clinical application. Large-scale, multicenter clinical trials will also be needed, particularly those focused on different populations based on race, gender, and age, to ensure the efficacy of exosome therapy across diverse groups.

Additionally, the application of exosomes involves issues such as cell sourcing, patient consent, and data privacy. Future research will focus on addressing these ethical concerns to ensure that exosome-based therapies comply with moral and legal requirements. As exosome technology advances, one of the key areas of future research will be finding ways to reduce the cost of exosome-based treatments, making them widely accessible and acceptable as a therapeutic option.

## 6. Conclusions

Exosomes containing biological contents can be transferred into the receipt cells, which strongly participate in skin flap survival through supporting angiogenesis and tissue regeneration. In particular, MSCs-derived exosomes such as the miRNAs like miR-23a and miR-126-3p, lncRNA like lnc-H19, as well as circ-RNA like circ29 have shown the clinical translation potential of skin flap survival. Future research should focus further on the large-scale production of exosomes, the establishment of standardized processing protocols, and the evaluation of clinical outcomes. With continuous technological advancements, exosomes are expected to become an important therapeutic tool in the future of medicine.

## Figures and Tables

**Figure 1 biomedicines-13-00353-f001:**
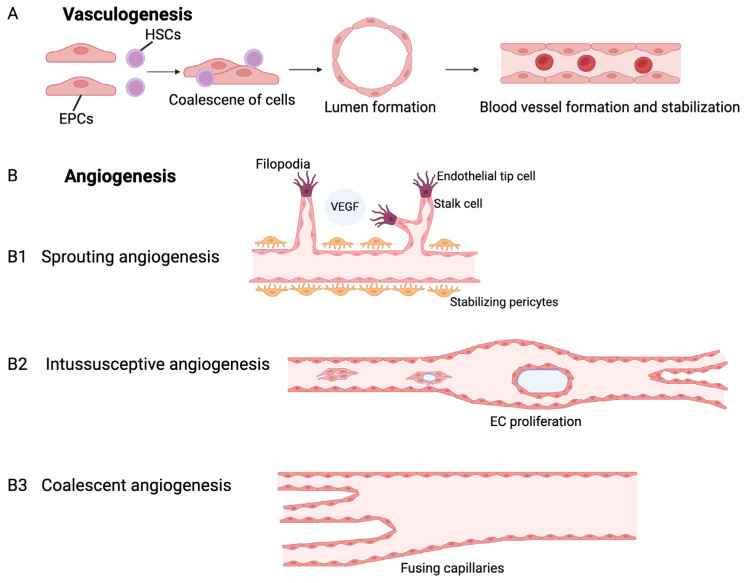
Vasculogenesis and the distinction among angiogenesis types. (**A**) Vasculo genesis is the process by which endothelial progenitor cells (EPCs) differentiate and aggregate to form the initial vascular network during early embryonic development, laying the foundation for the subsequent vascular system. (**B1**) Sprouting angiogenesis refers to the process in which endothelial cells (ECs) sprout from capillaries to form a vascular tree and establish a new capillary bed. In this process, endothelial tip cells play a crucial role by responding to the VEGF gradient, forming filopodia, and guiding the sprouting. Meanwhile, stalk cells proliferate to extend the new vascular branch. (**B2**) Intussusceptive angiogenesis is primarily mediated by ECs forming columnar structures between the opposing walls of capillaries. Under the stimulation of high levels of VEGF and other factors, ECs longitudinally divide a single vessel into two capillaries. This process expands the vascular bed, increases the number and complexity of blood vessels, thereby enhancing blood flow and oxygen supply to the local area. (**B3**) Coalescent angiogenesis can be regarded as the reverse of splitting angiogenesis. Blood vessels merge to form larger vessels, thereby improving circulatory efficiency. Abbreviations: EPCs (endothelial progenitor cells); HSCs (hematopoietic stem cells); VEGF (vascular endothelial growth factor); and EC (endothelial cell).

**Figure 2 biomedicines-13-00353-f002:**
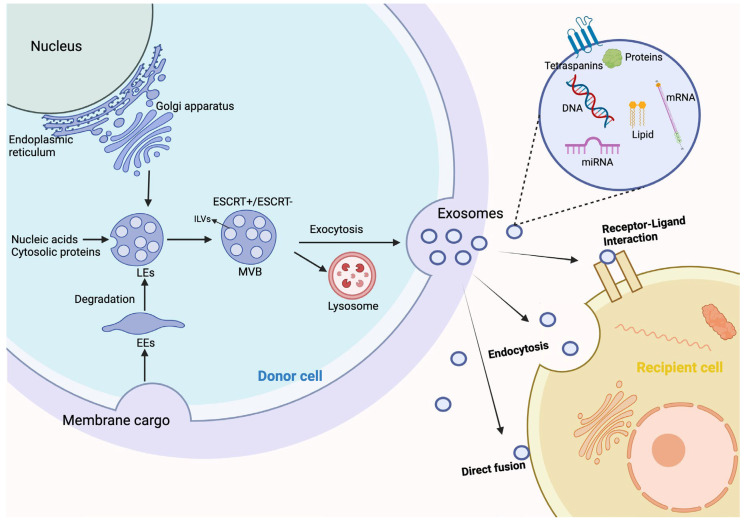
Exosome formation. Exosome formation starts with the creation of early-sorting endosomes via endocytosis. These endosomes mature into late-sorting endosomes, which generate intraluminal vesicles (ILVs) that will become exosomes. The late endosomes form multivesicular bodies (MVBs) that either release exosomes through exocytosis or fuse with lysosomes for degradation. The ESCRT (endosomal sorting complex required for transport) system plays a key role in this process, with ESCRT-III facilitating the inward budding and scission of vesicles. Exosomes can be taken up by recipient cells through endocytosis, receptor-ligand interaction, or direct fusion with the cell membrane. Abbreviations: EEs (early endosomes); LEs (late endosomes); MVBs (multivesicular bodies); ILVs (intraluminal vesicles); and ESCRT (the endosomal sorting complex required for transport).

**Table 1 biomedicines-13-00353-t001:** The mechanisms of exosomal miRNAs in angiogenesis.

Exosomal miRNAs	Origin	Target	Mechanism	Effect on Angiogenesis	Refs.
miR-23a	MSC	Sema6A, Sprouty2	Promotes endothelial migration and vessel formation	Pro-angiogenesis	[88]
miR-126-3p	hucMSC	PIK3R2	Activates PI3K/AKT/mTOR pathway, promotes endothelial growth	Pro-angiogenesis	[91]
miR-424-5p	Hypoxic Bone Marrow MSC	DLL4	Activates DLL4/Notch pathway, aids vessel formation	Pro-angiogenesis	[92]
miR-132	MSC	RASA1	Supports vessel formation by downregulating RASA1	Pro-angiogenesis	[93]
miR-16	Breast Cancer	VEGF	Inhibits VEGF, blocks blood vessel formation	Anti-angiogenesis	[94]
miR-100	MSC	mTOR/HIF-1α/VEGF	Inhibits vessel growth via mTOR/HIF-1α/VEGF signaling	Anti-angiogenesis	[95]
miR-133b-3p	Bone Marrow MSC	FBN1	Reduces oxidative stress, inhibits vessel formation	Anti-angiogenesis	[96]
miR-202-5p	High Glucose-treated ARPE19	TGFβR2	Inhibits EndoMT via TGFβR2, modulates TGF/Smad pathway	Anti-angiogenesis	[98]
miR-21-3p	MSC	PTEN, SPRY1	Enhances angiogenesis by inhibiting PTEN and SPRY1	Pro-angiogenesis	[99]
miR-214	Leukemia	Integrin α5	Promotes endothelial migration and tube formation	Pro-angiogenesis	[100]
miR-92a-3p	Retinoblastoma	KLF2	Increases angiogenesis by downregulating KLF2	Pro-angiogenesis	[102]
miR-21-5p	Colorectal Cancer	KRIT1	Activates β-catenin signaling, enhances angiogenesis	Pro-angiogenesis	[103]
miR-3157-3p	Non-small Cell Lung Cancer	TIMP/KLF2	Increases VEGF, MMPs, and permeability	Pro-angiogenesis	[104]
miR-320d	Colorectal Cancer	GNAI1	Triggers JAK2/STAT3, promotes vascular growth	Pro-angiogenesis	[105]
miR-210	Hepatocellular Carcinoma	SMAD4, STAT6	Promotes angiogenesis by downregulating SMAD4, STAT6	Pro-angiogenesis	[106]
miR-23a	Hypoxic Lung Cancer	PHD1/2, ZO-1	Enhances vascular permeability, promotes tumor vasculature	Pro-angiogenesis	[107]
miR-501-3p	M2 Macrophages	TGFBR3	Activates TGF-β pathway, promotes tumor progression	Pro-angiogenesis	[108]

**Table 2 biomedicines-13-00353-t002:** The mechanisms of exosomal lncRNAs in angiogenesis.

Exosomal lncRNAs	Origin	Target	Mechanism	Effect on Angiogenesis	Refs.
linc-CCAT2	Glioma	VEGFA, TGFβ	Upregulates VEGFA and TGFβ, promotes endothelial cell growth	Pro-angiogenesis	[111]
OIP5-AS1	Osteosarcoma	miR-153, ATG5	Inhibit the expression of miR-153, thereby upregulating the protein expression of ATG5	Pro-angiogenesis	[112]
H19	Liver Cancer (CD90+ Cells)	VEGF, HUVEC	Upregulate VEGF production and release in endothelial cells, enhance the ability of HUVEC cells to form tubular-like structures in vitro	Pro-angiogenesis	[113]
H19	Mesenchymal and Endothelial Cells	miR-106a, Angpt1	Acts as a sponge for miR-106a, upregulates Angpt1 expression, activates lnc-H19/Tie2-NO signaling pathway	Pro-angiogenesis	[115]
UCA1	Hypoxic Pancreatic Tumor	miR-96-5p, AMOTL2	Regulates miR-96-5p/AMOTL2 pathway	Pro-angiogenesis	[116]

**Table 3 biomedicines-13-00353-t003:** The mechanisms of exosomal circRNAs in angiogenesis.

Exosomal circRNAs	Origin	Target	Mechanism	Effect on Angiogenesis	Refs.
CircRNA-100338	Hepatocellular carcinoma	NOVA2	Activates mTOR pathway via NOVA2 interaction	Pro-angiogenesis	[117]
Circ29	Gastric cancer	VEGF	Sequesters miR-29a, activates VEGF pathway	Pro-angiogenesis	[118]
circSHKBP1	Tumor cells (specific origin not provided)	HUR	Sponges miR-582-3p, stabilizes VEGF mRNA	Pro-angiogenesis	[119]
cPWWP2A	Pericytes	Angiopoietin 1, Occludin, SIRT1	Sponges miR-579, regulates endothelial cell crosstalk	Pro-angiogenesis	[120]
Circ-Snhg11	ADSCs	NFE2L2, HIF1α	Activates miR-144-3p/NFE2L2/HIF1α pathway	Pro-angiogenesis	[121]

**Table 4 biomedicines-13-00353-t004:** The mechanisms of exosomal proteins in angiogenesis.

Exosomal Origin	Exosome Content (Protein)	Target	Mechanism	Effect on Angiogenesis	Refs.
Hypoxic stem cells of apical papilla (SCAPs)	HIF-1α, Jagged-1 (JAG1)	HUVECs	Activates HIF-1α/Notch1 pathway, increases VEGF	Pro-angiogenesis	[122]
Exosomes containing Annexin II (exo-Anx II)	Annexin II, tPA	Endothelial cells	Enhances tPA-dependent angiogenesis	Pro-angiogenesis	[123]
Exosomal DLL4	DLL4	Endothelial cells	Inhibits Notch, promotes tip cell formation	Pro-angiogenesis	[124]
Exosomes containing Tetraspanin Tspan8	Tetraspanin Tspan8	Endothelial cells	Enhances cell proliferation, migration, maturation	Pro-angiogenesis	[125]
Exosomes with latent TGF-β and betaglycan	TGF-β, betaglycan	Fibroblasts	Activates SMAD pathway, increases FGF2	Pro-angiogenesis	[126]
Exosomes enriched with Src, IGF-IR, and FAK	Src, IGF-IR, FAK	Prostate cancer cells	Activates Src/FAK signaling	Pro-angiogenesis	[127]
Exosomes from high-grade ovarian cancer	ATF2, MTA1	Endothelial cells	Activates VEGF/HIF-1 pathways	Pro-angiogenesis	[128]
Exosomes from high-grade bladder cancer urine	EDIL-3, EGFR	Bladder cancer cells	Activates EGFR signaling	Pro-angiogenesis	[129]

## Data Availability

No new data were created or analyzed in this study. Data sharing is not applicable to this article.

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
