# Peer review of "Exosomes in Skin Flap Survival: Unlocking Their Role in Angiogenesis and Tissue Regeneration"

_biomedicines, 2025, doi:10.3390/biomedicines13020353_

Round 1
Reviewer 1 Report
Comments and Suggestions for Authors
This manuscript is a comprehensive review of the role of exosomes in promoting angiogenesis and tissue regeneration, with a focus on their applications in skin flap survival. The structure is well-organized, the content is detailed, and the scientific foundation is robust. The manuscript effectively bridges the biological mechanisms of exosomes with their clinical potential, addressing a critical gap in the literature. Overall, it offers significant academic value and innovative perspectives in the cross-disciplinary area of exosome biology and reconstructive surgery. Below are detailed comments and suggestions for improvement:
First, the title is clear and accurately reflects the main focus of the review, making it appealing to the target audience. The abstract provides a concise and comprehensive summary of the background, key findings, and future directions. However, it could be further strengthened by emphasizing the unique advantages of exosomes in skin flap survival compared to traditional approaches, highlighting the manuscript's academic contributions.
The manuscript's structure is logical and well-organized, progressing from the fundamentals of exosome biogenesis and composition to their specific roles in angiogenesis and clinical applications. The background and current research are thoroughly discussed, providing readers with a solid understanding of the topic. However, the section on "Future Research Directions" is relatively brief. Expanding this section with a more in-depth discussion of challenges in clinical translation, such as the impact of exosome isolation and purification methods on therapeutic efficacy, standardization of treatment dosages, and long-term safety concerns, would enhance the manuscript's impact.
The figures are clear and enhance the manuscript’s readability by illustrating key mechanisms. However, some figure legends are overly brief. Providing more detailed explanations, especially about the relationships between the components shown in the figures and their roles in the discussed mechanisms, would improve clarity and accessibility.
The references are comprehensive and well-selected, covering both foundational studies and recent advancements in the field. They reflect the authors' thorough engagement with the literature. Nevertheless, adding brief critical evaluations of some key references, particularly those directly supporting the central arguments, would help integrate them more effectively into the review's narrative.
Finally, while the manuscript provides a strong scientific foundation, the discussion on clinical applications could be further enriched. Specifically, the challenges and potential solutions in applying exosome-based therapies for skin flap survival deserve more focus. Addressing issues such as optimizing exosome yield, ensuring uniform quality, and overcoming regulatory hurdles would make this section more insightful and actionable.
In conclusion, this is a high-quality manuscript with significant academic and clinical value. By addressing the above suggestions, particularly by expanding the discussion on future directions and clinical challenges, the manuscript can further enhance its scholarly contribution and practical relevance.
Author Response
|
Comments 1: The title is clear and accurately reflects the main focus of the review, making it appealing to the target audience. The abstract provides a concise and comprehensive summary of the background, key findings, and future directions. However, it could be further strengthened by emphasizing the unique advantages of exosomes in skin flap survival compared to traditional approaches, highlighting the manuscript's academic contributions. |
|
Response 1: Thank you for pointing this out. We agree with this comment. Therefore, we have made the revisions as suggested, which can be found on page 1 of the abstract from line 3, highlighted in red.
|
|
Comments 2: The manuscript's structure is logical and well-organized, progressing from the fundamentals of exosome biogenesis and composition to their specific roles in angiogenesis and clinical applications. The background and current research are thoroughly discussed, providing readers with a solid understanding of the topic. However, the section on "Future Research Directions" is relatively brief. Expanding this section with a more in-depth discussion of challenges in clinical translation, such as the impact of exosome isolation and purification methods on therapeutic efficacy, standardization of treatment dosages, and long-term safety concerns, would enhance the manuscript's impact. |
|
Response 2: Agreed. We have made the necessary revisions to emphasize this point, which can be found on pages 13-14, section 5.2, starting from line 5, highlighted in red.
Comments 3: The figures are clear and enhance the manuscript’s readability by illustrating key mechanisms. However, some figure legends are overly brief. Providing more detailed explanations, especially about the relationships between the components shown in the figures and their roles in the discussed mechanisms, would improve clarity and accessibility. Response 3: Thank you for pointing this out. We have made revisions to three figures and provided detailed descriptions for the figure legends. Additionally, we have created four tables to present the mechanisms more clearly.
Comments 4: The references are comprehensive and well-selected, covering both foundational studies and recent advancements in the field. They reflect the authors' thorough engagement with the literature. Nevertheless, adding brief critical evaluations of some key references, particularly those directly supporting the central arguments, would help integrate them more effectively into the review's narrative. Response 4: Thank you for pointing this out. Have revised as suggested, which can be found on page 5, 8, 9, 10, 11, 12, 13, 14, highlight in red.
Comments 5: Finally, while the manuscript provides a strong scientific foundation, the discussion on clinical applications could be further enriched. Specifically, the challenges and potential solutions in applying exosome-based therapies for skin flap survival deserve more focus. Addressing issues such as optimizing exosome yield, ensuring uniform quality, and overcoming regulatory hurdles would make this section more insightful and actionable. Response 5: Agreed. We have made the necessary revisions to emphasize this point, which can be found on pages 12-14, section 5.1-5.2, highlighted in red.
|

Reviewer 2 Report
Comments and Suggestions for Authors
Exosomes in Skin Flap Survival: Unlocking Their Role in Angi-ogenesis and Tissue Regeneration
In this review, Ji et al., present a very interesting topic where they explore the critical role of exosomes in promoting angiogenesis which is a key factor in skin flap survival. Skin flaps are commonly used in reconstructive surgery, and their survival
depends heavily on the formation of new blood vessels. It is important and timely topic since Exosomes, have emerged as vital players in intercellular communication, influencing
numerous biological processes, including angiogenesis. It is novel that this review delves into the molecular mechanisms by which exosomes facilitate angiogenesis. Additionally, it discusses their potential therapeutic applications in enhancing skin flap survival, and considers future research directions.
Despite these merits this review has certain flaws and I believe addressing these comments will certainly enhance the quality of this review. For example, there’s a sudden topic change to exosomes. I’d make it a subtle transition into the introduction.
Since this is a review article, I’d have a broader overview of the topic where I’d also mention bacterial counterparts of exosomes called Outer membrane vesicles or OMVs. Reference 11, where they mention the role of exosomes in cancer treatment I’d check the following article where employed bacterial exosomes or omvs for cancer treatment.
There’s a significant lack of referencing for example angiogenesis process, they didn’t cite it well or at all.
Please pay special attention to short forms and sign conventions for example they didn’t explain what is the full form of VEGF or NRP1?
The article does not flow as it is supposed to. I’d make concise and to the point avoiding jargons.
Check the sentence referenced with ref no 30. It sounds like contradictory ideas and I’d split the sentences which are too long. I’d also add line number so that it would have been easier to point to the sentences.
Long noncoding RNAs have been mentioned but it is beyond understanding why they didn’t mention about small non coding RNAs. Since this is a review I’d expect those in here.
I’d improve the visual presence by means of including more figures. I’ll make sure to have those figure more elaborate and self sufficient to reflect the content of the review.
The conscious drawn do not sufficiently describe the content of the review. I’d be more careful while describing the conclusions. Overall, this is an important review article in this area and I believe addressing these concerns will certainly help in advancing the field in this area.
I’d suggest the critical evaluation of this topic since it lacks the most important component of a review which critical evaluation and suggestions for future research progression.
Good luck and keep up the good work!
Author Response
|
Comments 1: For example, there’s a sudden topic change to exosomes. I’d make it a subtle transition into the introduction. Since this is a review article, I’d have a broader overview of the topic where I’d also mention bacterial counterparts of exosomes called Outer membrane vesicles or OMVs. Reference 11, where they mention the role of exosomes in cancer treatment I’d check the following article where employed bacterial exosomes or omvs for cancer treatment. |
|
Response 1: Thank you for pointing this out. We have made the revisions as suggested, which can be found on page 2 of the Introduction, highlighted in red. Reference 11, in the revised review version, is now cited as reference 10. This article does not mention the use of bacterial exosomes or OMVs for cancer treatment.
|
|
Comments 2: There’s a significant lack of referencing for example angiogenesis process, they didn’t cite it well or at all. |
|
Response 2: Agreed. We have made the necessary revisions to emphasize this point, which can be found on pages 5, section 2.3, highlighted in red.
Comments 3: Please pay special attention to short forms and sign conventions for example they didn’t explain what is the full form of VEGF or NRP1? Response 3: Thank you for pointing this out. We have reviewed all the abbreviations used in the article and provided their full forms.
Comments 4: The article does not flow as it is supposed to. I’d make concise and to the point avoiding jargons. Check the sentence referenced with ref no 30. It sounds like contradictory ideas and I’d split the sentences which are too long. I’d also add line number so that it would have been easier to point to the sentences. Response 4: Thank you for pointing this out. Have revised as suggested, highlight in red. Reference 30, in the revised review version, is now cited as reference 29 in page 4, have revised.
Comments 5: Long noncoding RNAs have been mentioned but it is beyond understanding why they didn’t mention about small non coding RNAs. Since this is a review I’d expect those in here. Response 5: We have discussed the small non coding RNA like miRNAs, circRNAs. We have revised in sections 4.1.1 to 4.1.4 on page 8-11.
Comments 6: I’d improve the visual presence by means of including more figures. I’ll make sure to have those figure more elaborate and self sufficient to reflect the content of the review. The conscious drawn do not sufficiently describe the content of the review. I’d be more careful while describing the conclusions. Overall, this is an important review article in this area and I believe addressing these concerns will certainly help in advancing the field in this area. Response 6: Based on revisions, we created four tables (Table 1-4) to present the mechanisms more clearly. Additionally, we have added a new figure regarding the types of angiogenesis (Figure 1) on page 3-4. And also provided detailed descriptions for the figure and table legends.
|

Reviewer 3 Report
Comments and Suggestions for Authors
The objectives of this study are valuable, and efforts have been made to achieve them. However, the following comments are suggested to enhance the quality of the study:
- Describe the conventional methods for enhancing angiogenesis in skin flap surgery and discuss their disadvantages and limitations in Section 2.
- Review the manuscript to ensure that all abbreviations (e.g., NRP1, TGFβ, ESCRT, EBV, BMSC, ADSC) are spelled out fully upon their first mention. This will assist readers who may not be familiar with these terms.
- Enhance the caption of Figure 1 by providing a more comprehensive description of the cargo and internalization routes into recipient cells.
- Improve the caption of Figure 2 by adding more detailed descriptions.
- Include the full names of abbreviations in the caption of Figure 3 to ensure clarity.
- Expand the discussion on the therapeutic potential of exosomes in enhancing skin flap survival, in alignment with the study’s title and objectives, by including further studies and interpretations.
Author Response
|
Comments 1: Describe the conventional methods for enhancing angiogenesis in skin flap surgery and discuss their disadvantages and limitations in Section 2. |
|
Response 1: Thank you for pointing this out. Have revised as suggested, which can be found on page 5, highlight in red.
|
|
Comments 2: Review the manuscript to ensure that all abbreviations (e.g., NRP1, TGFβ, ESCRT, EBV, BMSC, ADSC) are spelled out fully upon their first mention. This will assist readers who may not be familiar with these terms. |
|
Response 2: Thank you for pointing this out. We have reviewed all the abbreviations used in the article and provided their full forms.
Comments 3: Enhance the caption of Figure 1 by providing a more comprehensive description of the cargo and internalization routes into recipient cells. Response 3: Thank you for pointing this out. We have made revisions to Figure 1 (in the revised review version, is now Figure 2) and provided detailed descriptions for the figure legends in page 6, highlight in red.
Comments 4: Improve the caption of Figure 2 by adding more detailed descriptions. Response 4: Thank you for pointing this out. We have changed Figure 2 into four tables, which can more clearly describe the specific mechanisms. Each table has a detailed legend.
Comments 5: Include the full names of abbreviations in the caption of Figure 3 to ensure clarity. Response 5: Agreed. Have revised as suggested in page 11-12, highlight in red.
Comments 6: Expand the discussion on the therapeutic potential of exosomes in enhancing skin flap survival, in alignment with the study’s title and objectives, by including further studies and interpretations. Response 6: Agreed. We have made the necessary revisions to emphasize this point, which can be found on pages 12-15, section 5.1-5.2, highlighted in red. |

Round 2
Reviewer 3 Report
Comments and Suggestions for Authors
Accept